# The Roles and Challenges of Traditional Health Practitioners in Maternal Health Services in Rural Communities of Mthatha, South Africa

**DOI:** 10.3390/ijerph192013597

**Published:** 2022-10-20

**Authors:** Mvulakazi Patricia Thipanyane, Sibusiso Cyprian Nomatshila, Hannibal Tafadzwa Musarurwa, Olanrewaju Oladimeji

**Affiliations:** 1Department of Public Health, Preventive Medicine and Health Behavior Unit, Faculty of Health Sciences, Walter Sisulu University, Mthatha 5117, South Africa; 2Department of Human Biology, Faculty of Health Sciences, Walter Sisulu University, Mthatha 5117, South Africa; 3Department of Public Health, Community Medicine Unit, Faculty of Health Sciences, Walter Sisulu University, Mthatha 5117, South Africa

**Keywords:** traditional practitioners, maternal health, roles, challenges, rural

## Abstract

Traditional health practitioners (THPs) are considered as the entry level of care in African societies and play an important role in the delivery of health services to the population. A phenomenological qualitative study was carried out among purposefully selected THPs in Mthatha to understand their roles and the challenges they face in providing maternal health services. The study included a focus group discussion with seven participants which yielded three themes and seven subthemes. The content analysis of descriptive data from the focus group discussion revealed threats posed by unregistered and counterfeit THPs to the lives of pregnant women in rural settings. The THPs’ wide range of services allowed pregnant women to receive prenatal, antenatal, and postnatal care in proximity. However, this community level of care was marked by high levels of secrecy and counterfeit practitioners who used human body parts, which compromised the practice and rendered it unpopular. The South African government recommended the establishment of a register for THPs in order to protect the public, including the clarification of functional referral pathways between THPs and conventional health services.

## 1. Introduction

Traditional health services are differentiated by their approaches to re-establishing indigenous health models that existed prior to the advent of conventional medicine [1]. These services are reported to be used by most of the African population and are thought to play an important role in health care delivery, particularly in developing countries [2,3]. Traditional Health Practitioners (THPs) are said to play an important role in providing health services to the South African population, with responsibilities that include divination and herb distribution [4,5].

Literature has shown that THPs in various countries use a variety of traditional medicines for gynecological conditions such as stimulating fertility, inducing or stopping premature labor, and treating prolonged labor or placental expulsion [6]. Though the safety of using these traditional medicines is still being debated, traditional practices are said to play an important role for mothers and babies, and they are accessible, effective, affordable, and acceptable [7,8,9].

As part of divinity, religious leaders are said to heal pregnant women by praying and laying hands on them to reverse negative dreams about evil attacks on her and the baby [7]. Zerfu and colleagues document myths about evil spirits surrounding the death of the pregnant mother and the fetus in situations where undesirable health outcomes occur [10]. Because of indigenous beliefs and practices, some undesirable maternal outcomes are common among pregnant women who attend antenatal clinics late [11]. These beliefs included fear of bewitchment amongst pregnant women as the main reason to consult THPs for traditional medicinal products during pregnancy.

Inadequate functional integration of the traditional healthcare system and conventional medicine leads to the uncontrolled use of traditional medicines by the THPs, with traditional healthcare services perceived as a risk for conventional healthcare system implementation [12,13]. The THPs frequently express confidence in conventional medical professionals’ competence and respect for their practice. Mainstream physicians, on the other hand, do not have the same level of respect and trust in THPs, as evidenced by their refusal to refer patients to them for a variety of reasons, including traditional medicine’s lack of scientific foundation [14]. The THP’s holistic approach to patient care, which includes social, environmental, and spiritual well-being [15], contradicts this approach. Changes in attitudes and improved communication between the two health-care systems have been deemed beneficial for providing effective and efficient health-care services [16]. As a result of these approaches, THPs have earned the titles of custodians of traditional African religion and practices, cultural educators, counselors, facilitators, and spiritual protectors [5].

The South African government recognizes THPs as holistic care providers who provide health services to the population as well as holistic and comprehensive healthcare to patients [17]. The study explored THPs’ perceptions of their role in providing health services to pregnant women in South Africa’s Mthatha community in the OR Tambo District.

## 2. Materials and Methods

### 2.1. Study Design

The researchers used a qualitative phenomenological method to explore the role of the participants in the provision of traditional health services to pregnant women.

### 2.2. Setting

This study was conducted in the Mthatha, OR Tambo district South Africa. The area is located in the eastern part of the Eastern Cape province and has a population of 210,783 with an unemployment rate of 26% [18].

### 2.3. Population

The target population for the study consisted of traditional health practitioners (THPs) living in the Mthatha, OR Tambo district. It was estimated that in 2007 there were more than 10,000 THPs in the Eastern Cape province, where the OR Tambo district is located [19]. However, unpublished data reflects that there are more than 25,000 THPs in the Eastern Cape in 2022. These participants possessed vital information about traditional health practices and were regarded as custodians of the services. Due to the fact that there were few THPs who provided traditional services in the area, only seven THPs participated in one focus group discussion (FGD).

### 2.4. Sampling Procedure

Participants for the study were purposively enrolled using a snowball sampling approach. The focus group was held to explore the role of traditional health practitioners in providing traditional health practices and products to pregnant women in Mthatha, OR Tambo district. Mthatha is part of the OR Tambo district, which was designated as a National Health Insurance site by the South African government, as well as one of the districts with poor maternal health outcomes in 2013.

### 2.5. Data Collection

#### Focus Group Discussion

The FGDs were held at the public health facilities closer to the participants in November 2017. The THPs who took part in the FGD were given equal opportunities to participate, with no participant participating more or less than others. This allowed the discussion to progress to data collection that was equitable for all. Data saturation was determined by the absence of new information from participants after one hour and forty-five minutes. A qualified health professional with extensive experience in the clinical setting and in public health system management used an interview guide to facilitate discussions during the FGD by a public health and health promotion specialist. Table 1 shows an example of a question used during a discussion. Through the consent of the participants, all discussions during the FGD were facilitated in IsiXhosa and the audio was recorded. IsiXhosa is a native language of choice of the participants. To describe the social environment, non-verbal communication, participants’ comfortability, and specific behaviors, field notes were used. A professional transcriptionist then transcribed the recordings and documented them for interpretation.

### 2.6. Data Analysis

Data from the FGD were analyzed using content analysis. Using the content analysis method to ensure accuracy, transcription was meticulously executed through repeated stages of stop-rewind and repeat. A similar strategy was used to document the field notes, which were then combined with audio transcription. Professional language editors performed a forward and backward translation of transcriptions from isiXhosa to English and back to isiXhosa, ensuring consistency with original recordings. To confirm themes and sub-themes, an independent coder used a test and re-test approach. Following the completion of all data processes, all data in recordings was destroyed in accordance with confidentiality regulations.

### 2.7. Ethics and Consent

The Human Research Ethics and Biosafety Committee of Walter Sisulu University granted ethical approval with approval number 016/2016. The Eastern Cape province’s Department of Health granted permission to use health facilities under permit number EC 2017RP38 330. Participants gave written informed consent to participate in the FGD and have their audio recorded. A locker, an electronic coding system, a password-protected system, and further anonymization of the data were used to ensure the security of the collected data.

### 2.8. Measures for Trustworthiness and Reliability

Triangulation in the form of confirmability, transferability, credibility, and dependability was ensured in accordance with findings and recommendations from Guba et al. [20]. The researcher spent enough unhurried time with participants to ensure that they were adequately impressed by probing and asking follow-up questions.

## 3. Results

Data for this study was collected from seven THPs from Mthatha in the OR Tambo district who formed one FGD. Analyses from the data resulted in three themes and three subthemes, mainly under theme 3.

Table 2 indicates two themes and four sub-themes which emerged from the data. The first theme describes categories of THPs in the OR Tambo district and the second theme also has four subthemes.

### 3.1. Theme 1: Categories of Traditional Health Practitioners

Pregnant women received maternal services from various THPs, which included herbalists, diviners, and faith healers. The traditional health practitioners were asked about categories that existed in their practice and the following responses were obtained:


*“There are herbalists, diviners, faith healers. We are all working under the Department of Health.”*
Participant 1, THP-FGD


*“We are of different categories of traditional health practitioners; I will therefore ask Pastor’s wife to say something because I know her well when it comes to traditional healing”*
Participant 1, THP-FGD

### 3.2. Theme 2: Roles of THPs in Proving Traditional Health Services to Pregnant Women

#### 3.2.1. Pre-Conception and Antenatal Care

Although the services of the THPs is marked with high levels of secrecy of the ingredients used to prepare their concoctions, maternal services from prenatal to postnatal were reported. Participants reported that their services started when a young woman consulted them about enhancing their fertility and they provided them with a concoction for the purpose, and after using the concoction, pregnant women are reported to have presented to THPs with missed menstrual periods.


*‘We, traditional health practitioners, give a woman who has a problem of not conceiving a traditional mixture to drink”*
Participant 1, THP-FGD


*“It start s when a young woman starts attending antenatal care, she comes to us or traditional birth attendant when she has missed a period for one month after you gave her the traditional medicine for conception”*
Participant 3, THP-FGD

Medicines from THPs were reported to have functions for the prevention of involuntary abortions by pregnant women. This becomes essential in settings like Mthatha where health facilities are not in close proximity and pregnant women have to travel longer distances to receive care. This is essential for societies where a married woman would be measured by her ability to bear children.


*“We, traditional health practitioners know how to stop the occurrence of miscarriages using our medicinal products”*
Participant 4, THP-FGD

#### 3.2.2. Diagnosis, Treatment or Preventive Health

The strong belief systems of the Mthatha communities concerning the potential of evil spirits in tampering with pregnancies resulted in the need for pregnant women to seek protection for a successful pregnancy. Participants reflected on conditions they described as originating from evil spirits, which warranted specific traditional services for treatment and prevention. Walking at night and in dangerous places were considered by the participants as major risk factors for being exposed to evil spirits, and participants reported having natural powers to counter these. Their roles were reported to be in line with preventive health.


*“There is a certain herb that we use for protection of pregnancy from the evil spirits”*
Participant 7, THP-FGD


*“Another fact about the safety of pregnant woman is that there are dangerous places which a pregnant woman must avoid walking in them so that she is not affected by evil spirits. It is one of our responsibilities to protect them from these spirits by using some traditional medicinal products so that they do not have miscarriages”*
Participant 7, THP-FGD

#### 3.2.3. Management of Pregnancy Complication

Pregnant women often have different pregnancy-related conditions requiring consistent health support and care. When not managed properly and in a time-sensitive manner, pregnancy complications could result in maternal deaths and stillborn fetuses. Complications may include fluctuating blood pressure, swollen peripherals, a breech baby, and swollen feet. The THPs reported their abilities to manage these pregnancy complications through their secretive medicines, and maternal outcomes became desirable. Interventions for the management of complications included holy water, a mixture of herbs, castor oil and medicine for initiating sneezing. Therapeutic services such as the massaging of the stomach for the pregnant woman was also reported to have induced labor.


*“When pregnant women have swollen feet and legs, they come to us, and we treat them successfully”*
Participant 6, THP-FGD


*“People do not have the same problems when they are pregnant. There are pregnant women who present with swollen legs and feet, and they will be treated with either water traditionally prepared or prayed for, herbs, or traditional mixtures, because our traditional practices are different. I am a traditional health practitioner who uses most of these traditional medicines”*
Participant 2, THP-FGD


*“When I see that the woman is about to deliver and I notice that she is feeling cold and she tells me that she is in labour, I prepare a certain traditional mixture which I put in small bottle and thereafter I take Gin, mix it with little boiling water and give her to drink and tell her to go and sleep. When she reports that she is in labour, I switch over to Western medicine and she takes it together with a spoon of castor oil and she sleeps. If it happens that she goes to the hospital, she would deliver the baby well without operation”*
Participant 3, THP-FGD


*“When it comes to swollen feet of the woman who is about eight months pregnant, and the baby has turned but is not in the normal position, I take the medicine which I prepare from herbs which I am not going to explain the type of medicine; I use it to turn the baby to be delivered in the right position”*
Participant 1, THP-FGD


*“There is no need for a woman to go to the clinic or hospital when the baby is coming with legs first because we know how to correct when they come to us”*
Participants 2, THP-FGD


*“A woman will come to you with a breech presentation. I give her some stuff to make her sneeze, and then I assess her using my ‘water’ to massage her abdomen. Sometimes I mix the product that I use to make her sneeze with camphor body cream, petroleum jelly, and another confidential ingredient. I massage her abdomen, and the breech is corrected to the normal position”*
Participant 3, THP-FGD


*“If it happens that a pregnant woman comes to you reporting that she has delayed labour contractions I give her traditional mixture only and she gets contractions and delivers her baby well”*
Participant 7, THP-FGD

#### 3.2.4. Maintenance and Restoration of Health

While THPs play important roles in prenatal and antenatal care in Mthatha, they also play important roles in postnatal care. Poor postnatal care results in excessive bleeding when the placenta is retained and could ultimately lead to death. The restoration of health was reported as one of the key responsibilities THPs had, including the removal of retained placenta and explained processes involved. The procedure entailed inserting fingers into the vulva and blowing into a bottle until the placenta was ejected. While the placenta was successfully removed, the process’s sterility appeared to be compromised, with the potential for unintended infections.


*“When the woman has retained placenta retained placenta you tell her not to go to the hospital, and you attend to her. This is what we do, if there is a woman with retained placenta, you put your fingers in the vulva and pull it or we give her the bottle to blow until the retained placenta comes out”*
Participant 5, THP-FGD


*“It happens often that a woman comes to me having retained placenta and I remove it”*
Participant 3, THP-FGD

Every mother takes delight in delivering a healthy baby and is hopeful that the baby will remain healthy into adulthood. However, birthmarks, a dry tongue, an open mouth, and meconium were some of the concerns reported as commonly addressed by the THPs in their postnatal services. These concerns create worries for mothers and make their parenting frustrating. The THPs reported that they managed these concerns for babies using their mixture of traditional medicines and glycerin to restore and maintain health for newborn babies.


*“A baby with a natural birth mark will be having dry furred tongue and an open mouth. We, traditional health practitioners, mix glycerin with a certain type of traditional mixture. There are different types of traditional mixtures which we use to treat birth mark the baby, as well as meconium”*
Participant 1, THP-FGD

### 3.3. Theme 3: Challenges Faced by THPs in Proving Traditional Health Services to Pregnant Women

#### 3.3.1. Unregistered Traditional Practitioners

The government of South Africa requires that the THPs undergo accredited training and that their practices be regulated through the Interim Traditional Health Practitioners Council of South Africa. However, this process has been moving extremely slowly, such that even the THPs experience challenges from unregistered false THPs who expose their work to criticism and expose the public to scandalous operations like the use of human body parts during healing. This results in the deaths of innocent people. Pregnant women are also not spared in the process, as they fall victim to these unregistered THPs. Participants indicated that it was not the standard operation for THPs to use human body parts in the treatment of diseases, and often such practices are performed by bogus practitioners.


*“There are those fake traditional health practitioners who say they heal people and instead they cut the parts of the people”*
Participant 2, THP-FGD


*“There are no complications following the use of traditional medicine. Complications are seen when people consult other people who market themselves in town as traditional health practitioners by putting posts on the walls. People start undermining us and running after these so-called traditional health practitioners”*
Participant 2, THP-FGD


*“There are traditional health practitioners who are not working together with us, who do dirty things in the name of healing people”*
Participant 4, THP- FGD


*“Some traditional health practitioners say they can conduct deliveries telling lies instead they kill the innocent pregnant woman and her baby”*
Participant 6, THP- FGD

#### 3.3.2. Lack of Transparency about the Traditional Products Used by THPs

It is a common practice and an international standard that all medicines be approved and registered with relevant statutory bodies to ensure compliance and safety to the public. This is not the situation when it comes to traditional health practices whose operations are marked by an extensive lack of transparency on ingredients used to prepare medicines and processes involved in both preparations and healing. These are treated as individualized patents by the THPs. The participants who were traditional health practitioners reluctantly reported some of the practices and products they use to treat pregnant women.


*“We cannot as traditional health practitioners tell you, our secrets of how and which medicinal practices and products we use to treat pregnant women. We cannot give the government our secrets in case she steals our secrets. Otherwise, what we do, we go and dig the herbs, prepare them for the clients to be taken during the term of pregnancy. What we are confident of is that the woman delivers well without any complications”*
Participant 1, THP- FGD


*“… traditional health practitioners do not reveal their secrets, but if a person comes to us for consultation she goes back cured as if she was treated in the hospital”*
Participant 5, THP- FGD

#### 3.3.3. Concerns of THPs about the Protection of Their Practice

While the government developed regulatory frameworks for traditional health practices, the THPs expressed concerns about the protection of their practices and further protections of the public from the untrained and unknown THPs who often came from foreign countries. This situation was blamed on government’s failure to facilitate and strengthen their recognition, regulation, and constant monitoring and support for THPs. Traditional health services were viewed as a free for all and an easy way to make quick money by individuals who never underwent any training. which is required by government regulation. Due to a lack of proper training and initiation, these people mixed both wrong and unknown herbs, leading to adverse circumstances for pregnant women and their newborn babies. This was also reported to be a result of dissociation by these untrained THPs from the local association, which is responsible for information and experience sharing. This association was reported to further act as a contact point between local authorities and the THPs to facilitate training on emerging diseases and change protocols or guidelines for various conditions so as to improve performance, referral pathways, and role clarification.


*“The government is lagging behind in helping the traditional health practitioners towards authorisation of herbs, and as a result we do not want to mention the herbs we use which are very useful like a certain type of traditional medicine prepared from either the herb, stone or powder”*
Participant 6, THP-FGD


*“There are those who call themselves traditional health practitioners and yet they lie, those who have neither undergone any form of initiation nor have been called to treat people. Others do not want to be in our association”*
Participant 3, THP-FGD


*“There are people who say that they are traditional health practitioners and yet they do not know what they are doing, they bring complications to our people and tarnish our practice”*
Participant 2, THP-FGD


*“When the person gets complications, it will be said that it is us the THPs who caused the illness and yet it is these THPs from foreign countries. We usually ask ourselves what happened in their countries that they come and treat our people here because to them this is an employment opportunity”*
Participant 1, THP-FGD


*“Other traditional health practitioners do not know the herbs. They only heard about them, and they do not even know measurements and how they are prepared. This lack of skill would be seen when a woman gets complications which are a result of using the wrong medicine. This act brings bad image to our services as traditional health practitioners”*
Participant 3, THP-FGD


*“they are claiming that they cure birth mark by cutting it with a razor blade; and yet we know how to treat birth mark without cutting it. These false traditional health practitioners put on white beads to pose as trained traditional health practitioners*
*”*
Participant 5, THP-FGD

## 4. Discussion

This study shed light on the perceived roles and challenges that THPs face in providing traditional health services to pregnant women in Mthatha. Over the centuries, the African community has held a strong belief system that ranges from how a young female conducts herself before marriage to when she should marry and what should be done when pregnant or having a baby. This belief system has been central to how they care for and protect their children. Traditional health systems also played an important role in ensuring that women who wanted to conceive, pregnant women, and women with young children received services appropriate to their stage. These traditional health services, as opposed to the mainstream conventional health system, are frequently the first point of contact for people seeking health care, including pregnant women. Due to the limited access to conventional maternity care, African women frequently seek traditional medicine and services provided by THPs for reproductive and maternal health issues, as well as education on useful diets and successful breastfeeding [21,22,23]. THPs provide services such as strengthening the womb to improve its ability to conceive and carry the baby to term, protection against evil spirits that could endanger both the mother and the baby’s lives, management of the pregnancy with its associated risks and complications, the treatment of several postnatal conditions such as placental retention, and the treatment of neonatal conditions such as meconium removal. However, the diverse range of THPs providing services is beset by competing responsibilities and marred by challenges and controversies. Pregnant women would use a variety of traditional techniques and THPs based on their cultural and religious beliefs. Noting the importance of THPs in maternal health, a study conducted in Nigeria found that about a quarter of the population studied had their children delivered by THPs [22]. The current findings revealed that mothers were concerned about their own health as well as the health of their unborn or new babies, which led them to use THPs as soon as they were identified as necessary. Women were concerned about the exposure of their unborn and new babies to evil spirits, which could result in undesirable maternal outcomes. The primary motivations for using traditional medicines were reported to be concerns about the safety and health of the pregnant woman and the infant, protection from metaphysical forces, and socio-cultural beliefs [24]. Other studies [7,25] found that religious beliefs and divination played a role in decisions about alternative services to conventional health care. The most prominent THPs in the area, according to this study, are divine healers, herbalists, and faith-based healers. This was found to be consistent with previous research findings [26]. These practitioners were said to be capable of treating prenatal conditions such as infertility and complications in the first trimester. These prenatal conditions are common in many communities and are frequently blamed on unnatural causes. They are thought to be primarily managed by THPs. THPs were thus entrusted with the responsibility and authority to protect pregnant women from such dark forces. THPs have the ability to protect children from disease and evil spirits by fortifying the womb against witchcraft [9,27]. Findings from this study have shown that THPs use various culturally acceptable approaches to manage pregnancies and associated complications or problems, including unceremonious miscarriages, due to their abilities to diagnose, mix herbs, and pray. THPs, according to the literature, used traditional medicine to treat pregnancy-related issues [21]. Such abilities have also been linked to the prevention of involuntary abortions in pregnant women [27]. THPs reported that their skills ranged from protecting against evil spirits to preventing and managing pregnancy-related illnesses and complications. THPs reported swollen feet, breech, delayed labor, and colds as common problems seen and successfully treated by THPs, despite the lack of scientific evidence provided or recorded due to the secrecy of herbs used to treat various maternal ailments. The reported minimal side effects during their use, the rate at which these services are used, and the growth in the number of THPs seen in various communities all indicate confidence in the use of this level of care. While traditional health practitioners are frequently consulted as the first level of care in their communities, several studies show that they are also preferred to manage maternal difficulties, facilitate easy deliveries, cleanse the body after involuntary abortion, treat common colds, and treat a variety of other maternal ailments due to their minor side effects [28,29,30,31,32,33]. Furthermore, evidence has shown that THPs in South Africa are in charge of maintaining and restoring health [34]. Maternal health is divided into several stages that affect both the mother and the baby. After the mothers have given birth, it is critical that they remain under strict supervision for the first few days. This procedure ensures that any irregularities that may arise as a result of pregnancy and delivery are identified early on and that immediate action is taken to restore health. THPs reported being able to remove a retained placenta, treat newborns with furred tongues and open mouths, and treat birth marks. This finding corroborated the findings of a Taiwanese study of children, which discovered that traditional medicines were chosen for therapeutic purposes [35].The THPs reported several strengths in the provision of health services that were comparable to all other disciplines and service providers. Despite all of the THPs’ reported abilities and strengths, there were also reported challenges in practice. There were many unregistered and unqualified practitioners, according to the THPs, who made people do unexpected things and killed people. This included the use of human body parts in what the fake THPs claimed was healing. It was not anticipated that healing people would result in pain and death. The psychological impact of losing a family member, friend, or anyone else is a stressful experience that has frequently resulted in a tempered quality of life and reduced future prospects for all. Children may lose their parents, and parents may lose their children, all under the guise of healing by unregistered THPs. South African law prohibits anyone from working as a THP without being registered. However, the legislation’s implementation appeared hazy and unconfirmed. Participants in this study claimed that phony THPs were untrained, and were mostly from foreign countries with questionable health-care delivery methods. This was deemed detrimental to the THPs’ reputation. The general public is frequently unable to distinguish between trained and untrained, registered and unregistered THPs, leaving them vulnerable to unethical and potentially lethal practices. South African authorities condemned illegal practices and blamed them for poor maternal health outcomes [34,36]. The use of unregistered and unlicensed practitioners, combined with the practice’s secrecy, provided a fertile platform for phony THPs to operate, and the results were disastrous. According to the THPs, they were unable to reveal the ingredients used in the preparation of various remedies. Because there is no regulation on patents for their inventions and approaches, individual THPs are encouraged to keep their methods and herbs a secret. A patent would protect the THP from exploitation while also recognizing their work or skills. Patents would provide controls over the use of medicines or inventions and create ways for reporting side effects and adverse events, while also recommending ways to counteract or minimize the impact of those events in order to protect the public as end users. This method of maintaining secrecy reduces reliability and limits the hub of knowledge. Because of this level of secrecy, adverse outcomes after use are rarely recorded [25]. The THPs treated their curative art with reverence and secrecy in order to protect their patents, while others either refused to divulge their treatments or only issued an incomplete recipe [32,37,38].

## 5. Limitations

This was a qualitative study, and as such, its nature predisposes it to bias. The study only focused on a specific population of THPs, who constitute a minor community in society, exposing the study to limited coverage. There is a paucity of data on the number of available THPs offering antenatal services, thus limiting the generalization for the study. However, efforts were made to mitigate the effects of these constraints. Secrecy on ingredients and patents limited the study in terms of evaluating the scientific methods behind the traditional medicines and methods used by THPs.

## 6. Conclusions

Traditional health practitioners play a critical role in providing traditional health services to pregnant women in Mthatha’s King Sabata Dalindyebo sub-district. Although THPs provided essential prenatal, antenatal, and postnatal services to populations with limited access to public conventional facilities, the risks of services provided by unregistered THPs put pregnant women, mothers, and infants’ lives in jeopardy. Poor implementation of legislation governing traditional health practices jeopardized THPs’ practice and increased the risk of death, all while obscuring referral pathways between THPs and the conventional system. Extensive secrecy surrounding medications and approaches. However, the lack of disclosure of critical information about the products they used, as well as the preparation and administration of these traditional products, remained a challenge, especially when the products’ quality and safety were considered. The use of traditional and western medicines at the same time, as well as consultations with traditional health practitioners and health professionals, endangers the health of the pregnant woman and her unborn child, as well as the integration of health services in this sub-district. The services provided by untrained and illegal traditional health practitioners, which they frequently reported, raises concerns about the safety of their practice to pregnant women, particularly in terms of doxycycline. Managing drug or herbal intoxication, interaction, and reaction as a result of unregulated and mixed use of traditional medicines puts a strain on hospitals and the fiscus. The government must expedite registration, service regulation, and the implementation of a combined model of traditional and conventional health care. Community empowerment is also critical in identifying appropriate traditional service providers. A clear referral system must be developed and implemented that is recognized by both THPs and the conventional/mainstream health system.

## Figures and Tables

**Table 1 ijerph-19-13597-t001:** Discussion guide.

Area	Question
Knowledge	What are the different traditional health practitioners you know of?
	Which traditional services do you provide to pregnant women?
Traditional practices	How do you ascertain your competence of the service you provide to the pregnant women?
	How are your working relationships with other health providers?

**Table 2 ijerph-19-13597-t002:** Themes and sub-themes that emerged from the data.

Theme	Subthemes
1. Categories of Traditional Health Practitioners	1.1: Divine healers 1.2: Herbalists 1.3: Faith-based healers
2. Roles of THPs in proving traditional health services to pregnant women	2.1: Pre-conception and antenatal care
	2.2: Diagnosis, treatment or preventive health
	2.3: Management of pregnancy complications
	2.4: Maintenance and restoration of health
3. Challenges faced by THPs in proving traditional health services to pregnant women	3.1: Unregistered traditional practitioners
	3.2: Lack of transparency about the traditional products used by traditional health practitioners
	3.3: Concerns of traditional health practitioners about protection of their practice

## Data Availability

Data will be made available on in full appreciation of the Protection of Personal Information Act, and the confidentiality agreement with participants.

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
