# Peer review of "The Roles and Challenges of Traditional Health Practitioners in Maternal Health Services in Rural Communities of Mthatha, South Africa"

_ijerph, 2022, doi:10.3390/ijerph192013597_

Round 1

Reviewer 1 Report

Thank you for the opportunity to review the paper

Abstract:

Lines 28-30 are not clear. It would be better to rephrase the sentence.

Do THPs need to be registered in South Africa (Mthatha)? If yes, then Line 31 (THPs must be protected through registration needs to be rephrased.

It may be a good idea to provide more information on the study population. For example, the estimated number of THPs, the number of THPs providing antenatal services. The authors have reported it as "a few THPs". It may be worth clarifying it.

Section 2.3 Line 89... It is unclear if the authors want to represent seven participants as over-representation or under-representation of THP... (few THPs provide traditional services, and seven THPs participated)

Same with line 93 in section 2.4

The study period or the period of data collection is not clear in the manuscript. It needs to be reported.

Line 108... All audiotaped discussions were facilitated in IsiXhosa. What about discussions that were not audiotaped? Why was discussions partially audiotaped? The authors need to clarify this section.

Line 110... How did the authors ensure the reliability of data analysis?

Line 117... How were field notes documented with stages of stop-rewind and repeat? Does it mean that the interviews were video recorded not audio recorded? Please clarify. It is a major ethical issue. Authors need to be specific about the words used in the manuscript.

Line 155 needs to be removed (extra line)

The results section is described poorly. The authors need to elaborate on the themes and subthemes in their results. For example, in sub-theme 2.2... "participants reflected on conditions they described as originating from evil spirits".... it may be a good idea to elaborate what were the conditions that were described by the participants. 

It may be a good idea to discuss the scientific evidence on the practices of TBAs. For example, medications provided for the baby to turn (lines 203 -206).

As mentioned by one THP in the FGD that they have an association (Line 282-283), it may be a good idea to include background information on their association and how they are performing to improve the capacity building of the THPs in providing maternity care.

Discussion: In discussion, it would be a good idea to explore the outcomes of pregnancy treated by the THAs. Many of the practices they undertake may be potentially hazardous to the pregnant woman (for example use of medication to reverse breech in labour). Discussion should include both pros and cons of the practices. 

The conclusion section is extrapolated. The authors should provide a summary of their study in conclusion. 362-367

Reviewer 2 Report

The paper by Thipanyane et al titled "The roles and challenges of Traditional Health Practitioners in Maternal Health Services in rural communities of Mthatha, South Africa" reported  the challenges facing maternal health service providers in Mthatha, South Africa. 

On the whole, I think the manuscript is well written and the main conclusions are believable, I have only two minor suggestions for the authors.

1)  It is not entirely clear when the focus groups were interviewed, and this makes it hard to comprehend the broader temporal context. For instance, before COVID-19 would be a very different scenario compared to post-2019. The authors should make explicit the research design details.

2) The existing discussion is fairly weak, in the sense that it does not readily engage the broader literature on prenatal healthcare. The authors could elaborate on the practitioner and policy implications.
